# Monitoring of the Reproductive Cycle in Captive-Bred Female *Boa constrictor*: Preliminary Ultrasound Observations

**DOI:** 10.3390/ani11113069

**Published:** 2021-10-27

**Authors:** Mara Bertocchi, Enrico Bigliardi, Igor Pelizzone, Alessandro Vetere, Sabrina Manfredi, Diego Cattarossi, Matteo Rizzi, Francesco Di Ianni

**Affiliations:** 1Department of Veterinary Science, University of Parma, Strada del Taglio 10, 43126 Parma, Italy; mara.bertocchi@unipr.it (M.B.); enrico.bigliardi@unipr.it (E.B.); alessandro.vetere88@gmail.com (A.V.); matteo.rizzi@unipr.it (M.R.); francesco.diianni@unipr.it (F.D.I.); 2Ambulatorio Veterinario Belvedere, Via Pietro Bembo 12, 42123 Reggio Emilia, Italy; visc75@yahoo.it; 3Clinica Veterinaria Casale sul Sile, Via Massiego 4, 31032 Casale sul Sile, Italy; diegocattarossi@gmail.com

**Keywords:** *Boa constrictor*, reptile, reproduction, ultrasound

## Abstract

**Simple Summary:**

In recent years, reptiles have become increasingly popular pets. The growing interest in snakes has led to an increase in captive-bred ophidians, and the *Boa constrictor* is one of the most common reptiles bred in captivity. These snakes can be found in tropical South America, as well as some islands in the Caribbean. With the exception of one subspecies, the *Boa constrictor* is now included in Appendix II of the Convention on International Trade of Endangered Species (C.I.T.E.S.). In order to achieve the proper management of boas in captivity, it is essential to have in-depth knowledge on the reproduction of this species and to identify minimally invasive methods for a correct monitoring of the reproductive cycle and gestation, guaranteeing the animal’s welfare. Currently, ultrasonography is the most common technique used in veterinary medicine for evaluating reproductive activity in both mammals and reptiles. In this regard, knowledge of the *Boa constrictor* is rather scarce. A group of captive-bred female boa constrictors were monitored by ultrasound over an entire breeding season. Results suggest that this technique allows an accurate monitoring of the captive female boas’ reproductive cycle, as well as a precise control of the embryos’ development and viability.

**Abstract:**

The *Boa constrictor* is one of the most common reptiles bred in captivity. To achieve a successful breeding season, thorough knowledge of the females’ reproductive activity is necessary. In this regard, information on the *Boa constrictor* is still rather scarce. The aim of the present study was to monitor the ovarian activity and the embryonic development of boas by ultrasound. We performed brief scans on thirty non-anaesthetized snakes using a portable ultrasound system and a 7.5–10 MHz linear array transducer (Esaote MyLab™ Classic). Ultrasound features, dimensions, and echogenicity of the preovulatory and postovulatory follicles were determined. As gestation progresses, the postovulatory follicle size increases, and the embryonic silhouette becomes increasingly recognizable. During the second month after ovulation, by using color Doppler, early embryos’ heart activity could be evaluated. It is possible to highlight vascular connections between the mother and the membrane covering the embryonic structures. Ultrasound also allows one to identify follicular regression or slugs (nonfertilized eggs) early. The present study suggests that ultrasound could be an excellent noninvasive technique to evaluate the reproductive activity of *Boa constrictor*, allowing us to precisely identify the correct time for mating, monitor embryo development and viability, and allow the early diagnosis of follicular regression.

## 1. Introduction

*Boa constrictor* is one of the most common reptiles bred in captivity. A thorough knowledge of the reproductive activity of females and precise monitoring of embryonic development are important for successful breeding [1,2]. Although studies on ultrasound monitoring of the reproductive activity of different reptile species have been reported, it is important to study the characteristics of each species due to the existence of significant interspecies variability [3]. Boidae are a family of nonvenomous snakes primarily found in the Americas, although they also exist in Africa, Madagascar, Europe, Asia, and some Pacific Islands. For relatively primitive snakes, adults are medium to large in size, with females usually larger than males. The genus Boa includes two species and ten infraspecies [4]. In snakes, the female reproductive tract consists of long and slender ovaries located at two-thirds of the body coelom. The cranial part of the oviduct is dilated to form an infundibulum and then lines caudally along the intestine. The oviducts are opened to the cloaca separately as the vagina, without forming the uterus [5]. The vagina is opened into the cloaca through the urogenital opening or a separate opening into the cloaca [5]. In females, at the beginning of the breeding season, multiple follicles develop into Graafian follicles along with vitellogenesis to store yolk in ova [6]. After ovulation, the ova pass along through the infundibulum into the oviduct, and all ova are fertilized. In viviparous snakes, the embryo will develop in its individual fetal membrane within the oviduct [7]. Follicle-stimulating hormone (FSH) stimulates follicle growth and leads to production of vitellogenesis hormones, while luteinizing hormone (LH) stimulates ovulation and changes follicles to the corpus luteum to produce progesterone [8,9]. Most species of Boa, such as *Boa constrictor sp.,* are viviparous, with females giving birth to live young [7]. In the female’s body, individual membranes protect each embryo by regulating temperatures. Once born, the offspring have to break these membranes and get out of them. Viviparity has evolved many times within squamate reptiles, mostly in cool climates, but the selective advantages of the uterine retention of eggs remain obscure [10]. The independent origins of placentation have resulted in a variety of placental morphologies in different taxa, ranging from simple apposition of fetal and maternal tissues to endotheliochorial implantation that is homoplasious with mammalian placentation. In viviparous squamates, the functions of extraembryonic membranes found in oviparous species are retained or even enhanced [11,12]. In species with simple placentae, the chorioallantois remains a highly vascularized membrane that surrounds the embryonic hemisphere of the egg. Chorioallantoic capillaries are closely aligned with maternal capillaries, so gas exchange is still thought to be its primary function in viviparous squamates, although the transport of inorganic ions [11,13] and histotrophic transfer [14] are also functions. Ultrasonography is a very important diagnostic tool in any veterinary field, with many potential uses in reptile medicine [2,15]. This technique is considered safe for the patient; it is also a non-invasive method for the evaluation of anatomical position, size, and organ structure. The organs most suitable for ultrasonography examination in snakes are the heart, liver, gallbladder, kidneys, large intestine, and gonads [16]. This tool is also a suitable method for the observation of reproductive functionality, reproductive cycle, and reproductive-related pathologies [16]. In snakes, a lateral approach in ventral recumbency is excellent for assessing ovarian activity in females [17]. With regard to *Boa constrictor*, scientific contributions in this area are still rather scarce. This is a descriptive preliminary study which aims to evaluate the ovarian activity and embryonic development of captive *Boa constrictor* by BMode and Doppler ultrasonography.

## 2. Materials and Methods

The protocol of the study was approved by the “Organismo preposto al benessere degli animali (OPBA)” ethics committee of the University of Parma (n. prot. 06/CE/2019). All ultrasound sessions were performed on unsedated animals with minimal manual restraint, and all efforts were made to minimize stress.

The study was conducted at a professional breeding facility. Reproductive females were reared in standardized housing, handling, and environmental conditions. Each snake was individually housed in a rack with a dedusted beech chip substrate and a water bowl. In each rack, a probed thermostat allowed the temperature to be monitored, maintaining a gradient with a basking spot up to 35 °C. The relative humidity was kept between 50% and 70%, and a natural light/dark cycle (L/D cycle) was chosen.

Thirty adult *Boa constrictor* females were included in this study, 18 of which were pluriparous and 12 were primiparous. All were captive-born adults aged between 3 and 10 years, weighing between 7 and 16 kg and measuring from 2 to 3 m. The animals were fed every fifteen days with prekilled rats. Animals were monitored over one breeding season, starting from the beginning of the breeding season itself, from October to June. Data collected included medical history, coupling, ecdysis, and the presence of viable offspring. Upon ultrasound detection of follicles with a diameter between 18 and 30 mm [18], the female was housed with the male for a week. After the 7-day exposure to the male, ultrasound scans were performed weekly until the gestation term (approximately 100–110 days after post-ovulatory shed [18]). For each female, brief lateral scans of the lower half of the body were performed weekly after applying a layer of conductive gel. Each scan was performed on unsedated animals kept in sternal and lateral recumbency. Lateral and ventral scans of the medium and caudal third of the body were obtained in accordance with Hochleitner and Hochleitner’s [19] and Schilliger’s [17] techniques. The procedure was carried out on the farm using a portable ultrasound system equipped with a 7.5–10 MHz linear array transducer (Esaote MyLabTM ClassicC^®^, Genova, Italy). Settings for color and power Doppler were constant for all examinations (color gain 60%, medium wall filter, pulse repetition frequency 700 Hz, image depth 5–7 cm). The ultrasonographic examination was performed by the same experienced operator. Depth and contrast were adjusted to optimize the structure display. Ultrasound features included the dimensions and echogenicity of the ovarian follicles and the development and viability of the embryonic structures, as well as any cases of follicular regression and the evaluation of possible nonfertilized, yellowish eggs, known as slugs [15]. The viability of the embryonic structures was studied using color and power Doppler. The ultrasound session data were reported in tables, including the date, the largest follicular diameter in mm, the sonographic appearance of the observed structures, and any additional notes, such as skin shedding and mating. Using an Excel spreadsheet, time intervals (in days) were calculated for the most significant reproductive events: the onset of ultrasonographically detectable embryonic cardiac activity, skin shedding, and parturition. The mean duration of gestation starting from the post-ovulatory shed was evaluated. The gestation durations of primiparous and pluriparous females were then separately determined. Data were tested for normality through the Shapiro–Wilk Normality Test. Since they were normally distributed and having to compare two quantitative variables required studying two groups, Student’s *t*-test was performed to determine the significance, which was set at *p* ≤ 0.05 [20]. 

## 3. Results

Regarding the length of gestation starting from the post-ovulatory shed, an average duration of 101.1 days was found, with a standard deviation (SD) of 4 days. Considering mean values and SD of primiparous and pluriparous females separately, an average gestation of 101.8 ± 4.9 days and 100.7 ± 3.8 days was calculated, respectively. However, the difference was not statistically significant (*p* > 0.05). Considering the ovulation as a reference point, the average gestation was 124.5 ± 7 days. Distinguishing also in this case primiparous and pluriparous, mean lengths equal to 125 ± 5 and 124.2 ± 8.5 days were highlighted, respectively—nevertheless, with no statistically significant difference (*p* > 0.05). On average, post-ovulatory shed was observed 24.6 ± 7 days after ovulation. The pluriparous ones, in particular, showed the post-ovulatory shed after 25.5 ± 4.8 days after the onset of the ovulatory swelling, while the primiparous was 18 ± 7.1 days, but this was not a statistically significant difference (*p* > 0.05).

Regarding the ultrasound aspect of the structures, no differences appeared between pluriparous and primiparous animals.

Ovarian follicles were highlighted on both sides within the caudal half of the females’ body with a ventrolateral approach. Before ovulation, numerous rounded follicles could be recognized. Up to a diameter of 10–18 mm, these structures are anechoic and rounded in size, arranged in clusters in the ovarian stroma, or aligned in a chain arrangement (Figure 1).

A full 100% (30/30) of the boas coupled. When the diameter reached at least 18 mm, the female accepted the male, who was housed in the rack with her for a week (until the follicles had a maximum diameter of 30 mm). At this point, the follicles had a less uniform ultrasound appearance, and the shape became oval (Figure 2).

As the diameter increased, follicles indeed show areas of different echogenicity: all females showed follicles with a central anechoic “nucleus” surrounded by a more echogenic peripheral area (Figure 3).

In 90% of females (27/30), follicles with an onion-ring-like appearance were highlighted (Figure 4).

The follicular diameter increased to over 40 mm at ovulation, immediately after which a slight decrease in size (a few mm) could be detected. This phase was characterized by the swelling of the females caudal third and was recognized in 83.3% of cases (25/30). In all these specimens, follicles were oval and of nonuniform echogenicity, with anechoic areas alongside areas of greater echogenicity (Figure 5).

In the first month of the postovulatory phase, the period in which the postovulatory shed occurs (2–3 weeks after ovulation), the shape of the follicle was oval, and all females who have given birth to living and viable offspring presented postovulatory follicles with a heterogeneous appearance, characterized by hyperechoic and anechoic areas; in particular, an anechoic central area was clearly visible, and in 96.2% of cases, follicles with an onion ring appearance have been highlighted (Figure 6, Figure 7, Figure 8). In this phase, however, 19% of the animals that gave birth to live snakes (5/26) also presented follicles with a more homogeneous widespread echostructure. Females with these structures then laid one or more slugs (nonfertilized eggs).

In 88.5% of subjects who carried the pregnancy to term (23/26), a small roundish anechoic area was observed at the edge of the follicles, apparently in relation to the external membrane of the follicles themselves (Figure 9a,b).

During the second month after ovulation, the embryonic vesicle could be recognized on the sonogram, and initial cardiac activity could be observed.

In particular, embryo vascularization could be assessed using color Doppler (Figure 10). In B-Mode, the embryo appeared as an anechoic, rounded peripheral area, while the remaining part of the egg being formed was more echogenic. With development, eggs increased in size, and the area around the vesicle became mostly anechoic, except the outermost area, which remained hyperechoic (Figure 11). Between 35 and 45 days after ovulation, early embryonic heart activity could be measured in 100% of females who gave birth to live and viable offspring (Figure 12). In this period, a median systolic peak velocity of 0.16 m/s was calculated, while the median diastolic peak velocity was 0.05 m/s. As the deposition time approached, embryonic development, vascularization, and viability were monitored.

In all pregnant females, through the use of Doppler, it was also possible to highlight vascular connections between the mother and the membrane covering the embryonic structures in this phase (Figure 13).

In the second third of the gestation period, in 100% of females who have completed gestation, the embryonic vesicle size increased and embryonic development led to the appearance of a “snail” structure, reflecting the coiled look of the embryos (Figure 14).

In the last third of gestation, the fetuses grew to occupy most of the available space, taking on a “spiral” appearance, recognized in all females who have given birth to live offspring (26/26). The ribs were easily recognizable, and embryo vitality could be assessed thanks to the examination of vascularization in spiral-looking fetuses, alongside the movements of the fetuses themselves (Figure 15).

During the last weeks before delivery, progressively more significant fetal movements were observed in all females (26/26). In this phase, the fetus’s position was quite variable. The size of fetuses increased, and the skeleton was clearly visible (Figure 16). Moreover, during the last weeks preceding the birth, it was possible to assess fetal heart activity in more detail in all pregnant boas. The cardiac morphology of the fetuses has been evaluated (Figure 17).

Four out of thirty animals (two pluriparous and two primiparous) showed follicular regression. Follicles over 30 mm in diameter regressing to an appearance characteristic of the previous stages of development were observed, with a decrease in diameter and changes in echogenicity.

## 4. Discussion

Professional reptile breeding is becoming increasingly important, and it is therefore particularly useful to develop reliable techniques to monitor reproduction and gestation, maximizing the potential of the breeding itself [2,15,21,22]. In the present study, behavioral data were integrated with ultrasonography, a noninvasive diagnostic technique, minimizing animal stress. In reptile ultrasound, the presence of air trapped under the scales could cause artefacts [17,23]. In this study, this was overcome by applying a copious layer of ultrasound gel to the snake and probe, allowing a clear visualization of the coelomic structures [2,23]. Rib shadowing can also create confusion by mimicking the presence of follicles near large abdominal vessels, and small abdominal blood vessels could mimic the presence of small follicles [23]. In this regard, the use of color Doppler could be useful to clearly recognize follicles, as confirmed by the present study [2,17,23]. The reproductive structures of ophidians can undergo significant shifts in relation to the ingestion of a meal [24]. In our case, this interference was not observed, and meals did not prevent adequate ultrasonographic highlighting of the reproductive structures. As with other snake species, it also seems for *Boa constrictor* that the best approach for follicle visualization is ventrolateral visualization [2,17,19]. From this preliminary study, it emerged that, in the preovulatory phase of the reproductive cycle, the ovaries were characterized by small, rounded, and numerous previtellogenic follicles. This is in agreement with what is reported in the literature for *Boa constrictor* and Royal python [2,15,18]. In *Python regius,* roundish and anechoic follicles have been highlighted before mating, although they are on average smaller in size [2]. The follicle arrangement observed in the considered boas has also been reported by studies carried out on other species of snakes [2,25].

When the follicles have a diameter of 17 mm, mating takes place. The chances of fertilization remain until the follicles reach a diameter of 30 mm [18]. Therefore, thanks to ultrasound monitoring, it is possible to precisely identify the correct moment for inserting the male into the rack in view of mating. Follicles then appear more irregular, with anechoic areas alongside areas of greater echogenicity. Follicles with an onion ring appearance can also be observed. This increase in echogenicity is compatible with the vitellogenic phase of folliculogenesis, characterized by the deposition of lipids and proteins inside the follicle, which increases in size [18]. A certain variability among individuals emerges from ours. The existence of both interspecific and intraspecific variability in the reproduction of reptiles is indeed documented in the literature [26,27,28]. When ovulation occurs, with interindividual variability, swelling of the caudal third of the snake is highlighted [18,21]. This is confirmed by the present study, in which the phenomenon was recognized in several subjects. Post-ovulatory follicles have a more oval shape than pre-ovulatory follicles [18]. At the time of birth, in addition to newborn snakes, slugs are often present [15]. Our research confirms this aspect. From an ultrasound point of view, in some species, a difference between fertile eggs and slugs can be appreciated in the early stages of gestation, as described in Royal python [2]. In the case of *Boa constrictor*, from this preliminary study, it seems that it could be possible to identify and differentiate such structures early. Our hypothesis and observations are that, in the first month after ovulation, slugs are more echogenic than the fertilized structures, which seem to have a more hypoechoic central area. Further studies are needed with a major number of subjects to elaborate on these findings.

Currently, ultrasonography is one of the most common techniques used in veterinary medicine for evaluating reproductive activity in both mammals and reptiles. In different species of reptiles, ultrasonography has also been recognized as a valid technique for assessing embryonic vitality and development [2,15,17,18,19,29,30,31]. This seems to be confirmed for *Boa constrictor* too. During the second month after ovulation, the embryonic vesicle could indeed be highlighted, and the initial cardiac activity could be observed. This confirms what was observed in two red-tailed boas (*Boa constrictor constrictor*) and three Amazon tree boas (*C. hortulanus*) [15]. As in the Royal python, embryo vascularization could be observed by color Doppler in *Boa constrictor*. The present study also shows the possibility of measuring the cardiac activity of embryos, in particular considering systolic and diastolic peaks, as well as heart frequency. As already shown for other species, this is particularly useful for monitoring the vitality of embryos during gestation [2]. Regarding viviparous species, the exchange of materials between the mother and fetus through contiguous and highly vascularized membranous surfaces has been reported [7,32]. This is confirmed in our study. In fact, important vascular connections between mother and fetuses were detected using color Doppler. The size of the embryos progressively increased, and, in the second half of the gestation period, the silhouette of the embryos could be clearly highlighted. Their shape became better defined, passing from a snail-like appearance to a spiral shape. This snail-like ultrasonographic aspect recalls the morphology described in relation to the embryos of *Elaphe quadrivirgata*, an egg-laying species isolated from just-laid eggs [33]. A similar morphology was also macroscopically described in the case of the common garter snake (*Thamnophis sirtalis sirtalis*), *Python sebae,* and Japanese mamushin (*Gloydius blomhoffii*) embryos [34,35,36]. In *Python sebae*, embryos showed a decreasing number of coils as development progressed [35]. Our study shows that when boa embryos show a spiral appearance, the ribs are easily recognizable, and embryo vitality can be assessed thanks to the movements of the embryos themselves and to the examination of vascularization. Moreover, in the days preceding birth, it is possible to evaluate the cardiac morphology of the fetuses. Four cases of follicular regression were observed, in particular two primiparous and two pluriparous females. It is possible to distinguish this phenomenon from physiological follicular atresia, considered the normal reduction of the initial high number of follicles during the reproductive cycle [23,37,38]. Follicular regression is instead intended as the massive reabsorption of follicles in advanced stages of development, with the consequent return of the female to an earlier phase of the reproductive cycle, up to the development of new follicles [2]. The findings from this study are in line with the literature. Cases of follicular regression have indeed been reported for other snake species, such as Royal python [2,23,39]. In *Python brongersmai,* a correlation between follicular regression and the presence or absence of males during follicular development has been suggested, but the causes of the phenomenon have not yet been clarified [39]. Furthermore, it is not yet clear whether there is a correlation between follicular regression and primiparous or pluriparous animals. The authors intend to investigate this aspect in a subsequent study, involving a greater number of animals.

In this study, the mean gestation length counting from the post-ovulatory shed was equal to 101.0 ± 4 days, and 124.5 ± 7 days after ovulation, which is quite consistent with the literature. Boa constrictors give birth approximately 105 days after postovulatory shedding and approximately 123 days after ovulation [1,21]. In comparing the mean gestation length of primiparous and pluriparous, there were no significant differences. Females considered in our study showed a post-ovulatory shed about three weeks after ovulation. This confirms what is reported in the literature. Indeed, pregnant *Boa constrictor* females generally exhibit post-ovulatory shed approximately 16–20 days after ovulation [1,21].

## 5. Conclusions

Ultrasonography is one of the most common techniques used in veterinary medicine for evaluating normal and pathological aspects of the reproductive system in both mammals and reptiles [2,17,19,29,40]. The present study confirms previous findings that the reproductive cycle of a female snake can be precisely followed by ultrasonography [2,15,23,41]. This technique also allows one to monitor follicular development and highlight the optimal reproduction time. To date, some studies relating to the ultrasonographic evaluation of *Boa constrictor* have been reported in the literature [7,11,15]. However, those dedicated specifically to the reproductive structures of this species are very few, and above all, a small number of subjects is considered. The present study examines a greater number of *Boa constrictor* females, with specific ultrasound evaluation of ovarian activity as a whole. The most innovative aspect is the monitoring of pregnancy not only with BMode ultrasonography, but also with the use of Doppler. Indeed, in the present study embryonic blood flows and embryonic cardiac activity were highlighted. This result is essential for monitoring embryonic development and viability during gestation.

Moreover, this technique could also be useful in this species to early identify slugs and follicular regression cases. 

## Figures and Tables

**Figure 1 animals-11-03069-f001:**
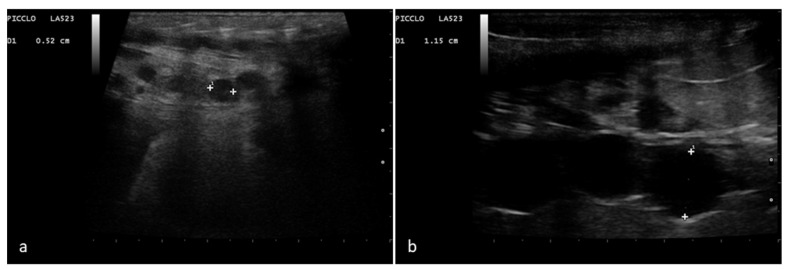
Rounded and aligned anechoic pre-ovulatory follicles of different sizes in a female *Boa constrictor*. Phase recognized in 100% of the subjects involved in the study. (**a**) Boa with 5 mm diameter follicles, before ovulation. (**b**) Boa with 11.5 mm diameter follicles, before ovulation.

**Figure 2 animals-11-03069-f002:**
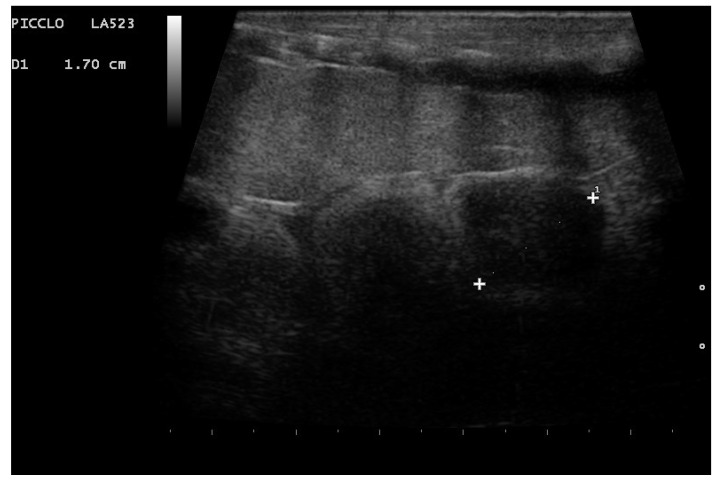
Follicles 17 mm in diameter in a female boa at the beginning of the mating period. Phase recognized in 100% of the subjects involved in the study. The structures no longer appear round and uniformly anechoic but show a nonuniform echogenicity.

**Figure 3 animals-11-03069-f003:**
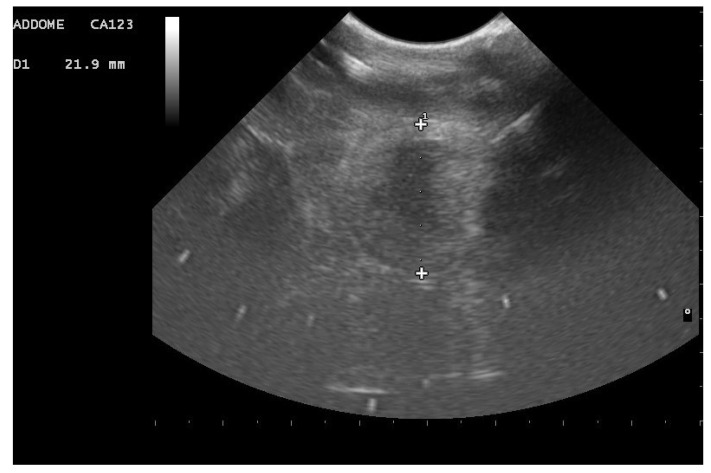
Follicle in a female boa during coupling. The follicle shows a central anechoic area surrounded by a more echogenic peripheral area. Recognized in 100% of the females involved in the study.

**Figure 4 animals-11-03069-f004:**
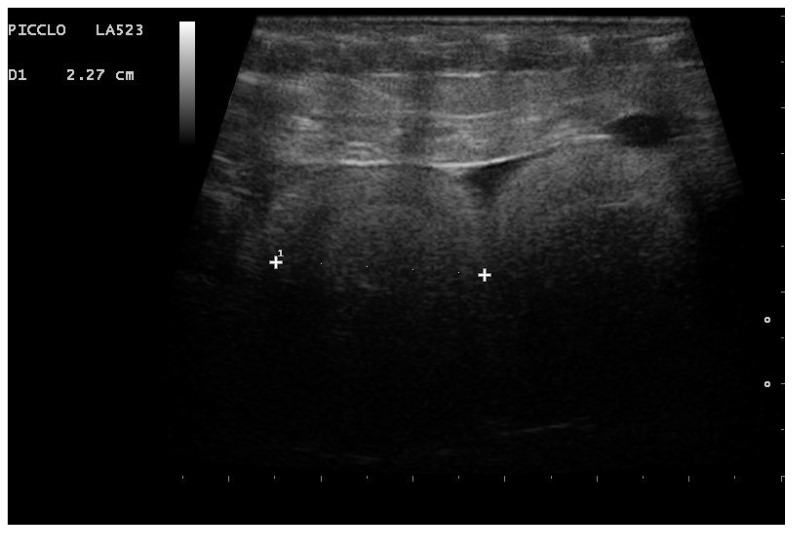
Follicle with an onion-ring appearance in a female during coupling. A concentric ultrasound appearance was recognizable in 90% of females.

**Figure 5 animals-11-03069-f005:**
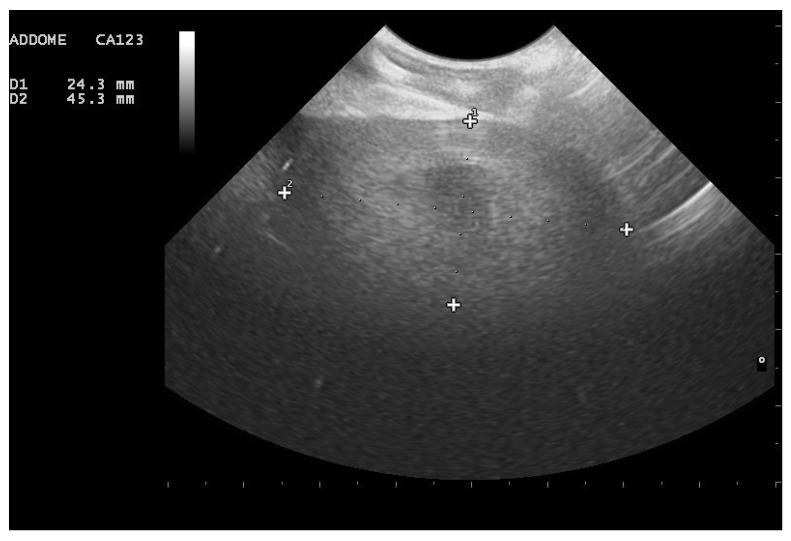
Follicle in a female boa during ovulation. An oval shape with a diameter greater than 45 mm is observed in 100% of females in whom swelling was recognized at ovulation. The central part has lower echogenicity than the peripheral part.

**Figure 6 animals-11-03069-f006:**
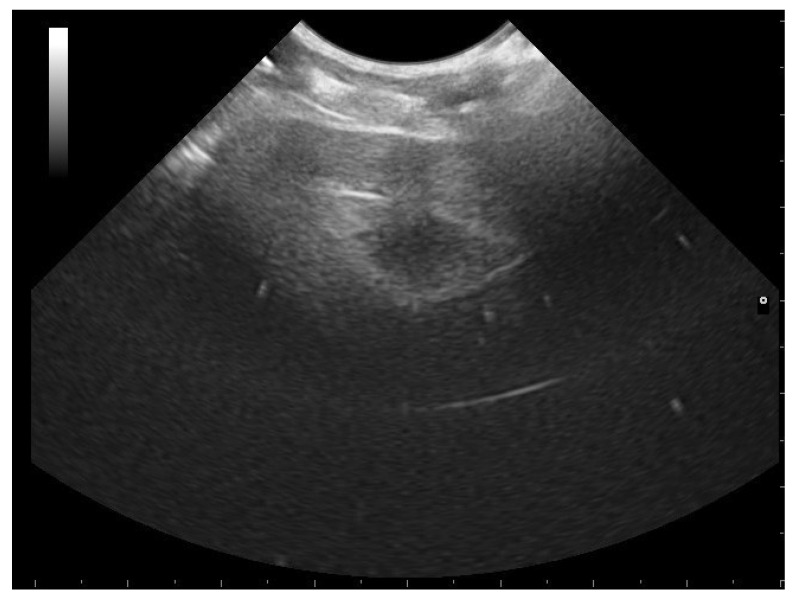
Follicle highlighted in a female boa 23 days after ovulation. An oval shape is shown, and the central anechoic area is highlightable in all females who have given birth to living and viable offspring.

**Figure 7 animals-11-03069-f007:**
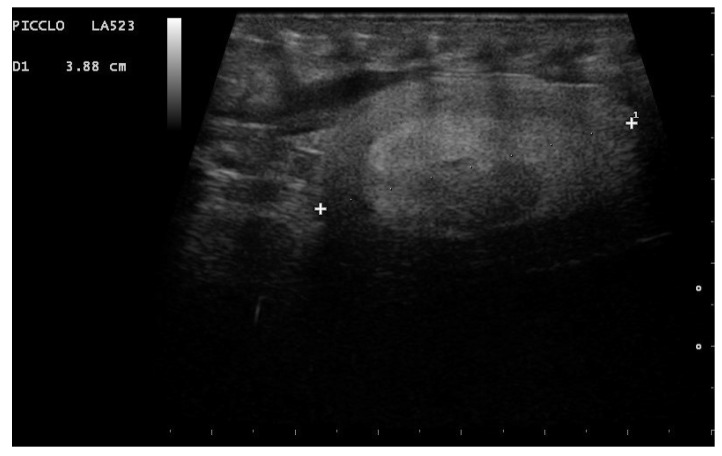
Follicle highlighted in a female boa 24 days after ovulation. Due to the alternation of areas with different echogenicity, an onion ring appearance is highlighted in 96.2% of females who have given birth to living and viable offspring.

**Figure 8 animals-11-03069-f008:**
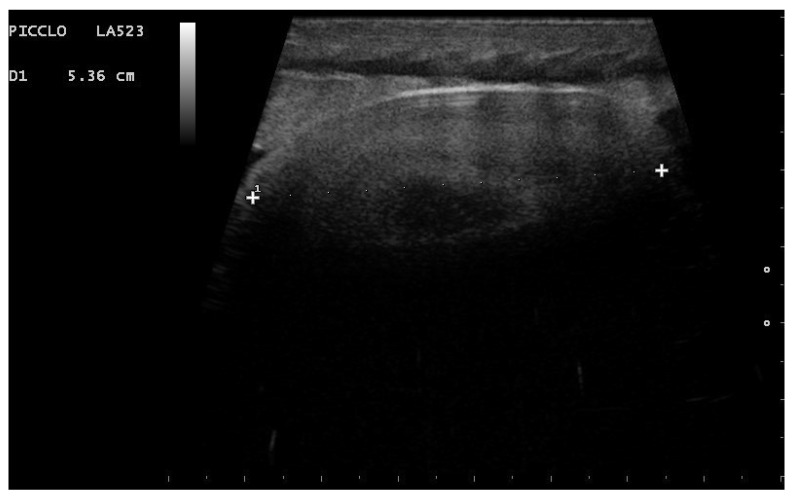
Post-ovulatory follicle highlighted in a female boa (32 days after ovulation). An onion ring appearance and an anechoic central area have been observed in 96.2% and 100%, respectively, of females who have given birth to living and viable offspring.

**Figure 9 animals-11-03069-f009:**
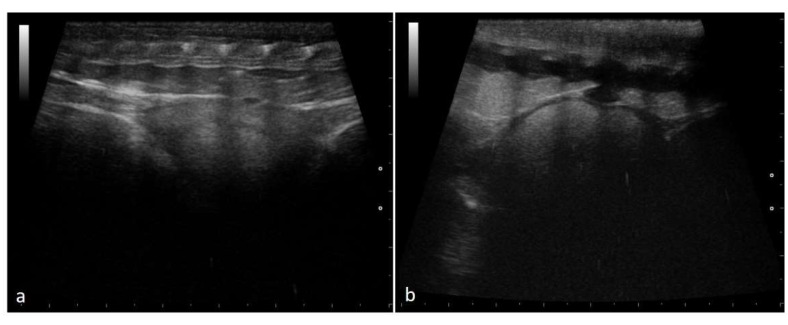
Postovulatory follicles in a female boa at 28 days (**a**) and 32 days (**b**) after ovulation. A small rounded anechoic peripheral formation was highlighted in 88.5% of subjects who carried the pregnancy to term.

**Figure 10 animals-11-03069-f010:**
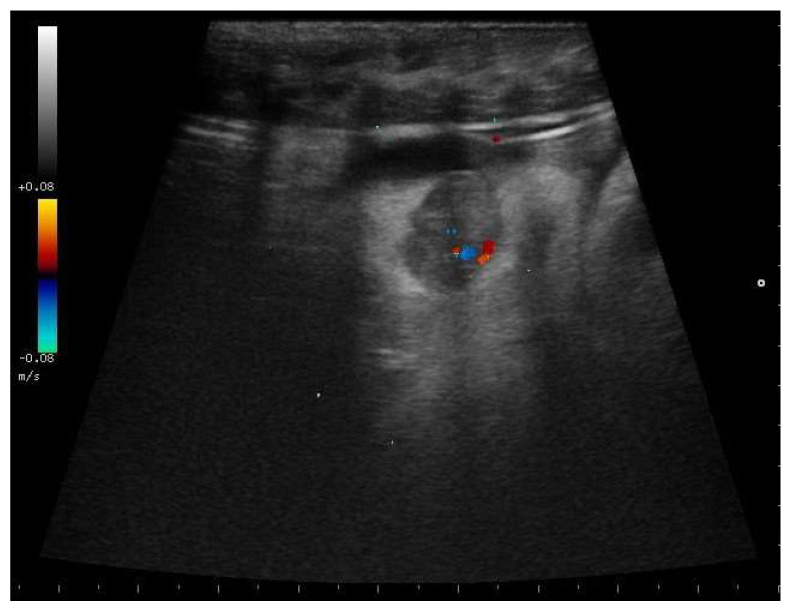
Gravid female boa in the first month after ovulation—flow towards the transducer is depicted in red, while flow away from the transducer is shown in blue. In 100% of females who gave birth to live and viable offspring, it was possible to observe embryonic vesicle, and blood flow was highlighted by color Doppler.

**Figure 11 animals-11-03069-f011:**
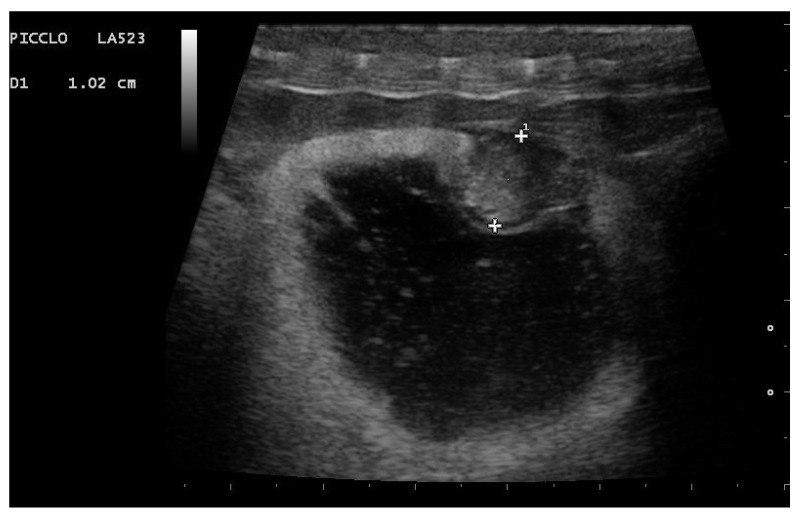
Gravid female boa 42 days after ovulation. An embryonic vesicle 10.2 mm in diameter, surrounded by a large anechoic area, is clearly visible. The outer membrane is instead hyperechoic. Phase recognized in 100% of females who gave birth to live and viable offspring.

**Figure 12 animals-11-03069-f012:**
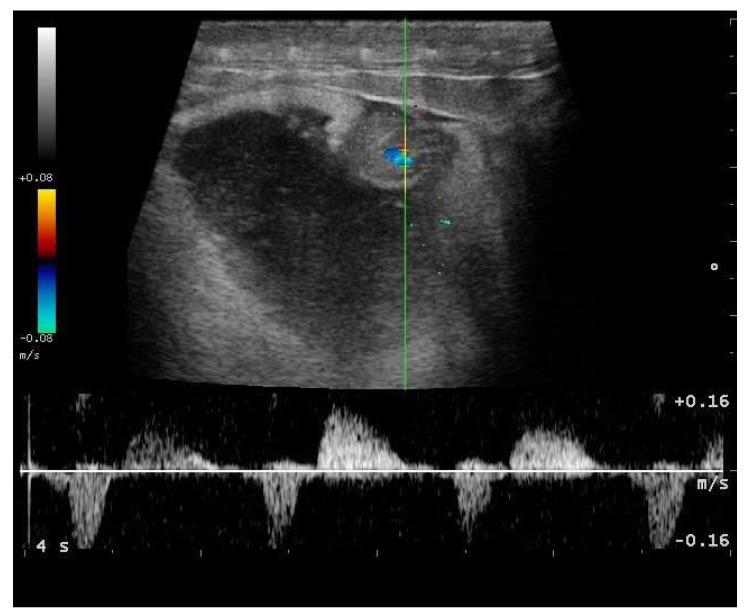
Gravid female boa 42 days after ovulation—flow towards the transducer is depicted in red, while flow away from the transducer is shown in blue. Through Doppler, it was possible to highlight embryonic blood flows and heart activity in 100% of females who gave birth to live and viable offspring.

**Figure 13 animals-11-03069-f013:**
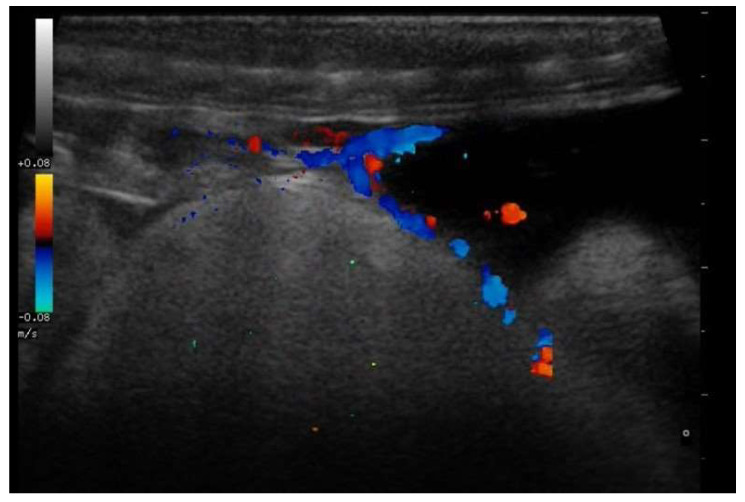
Vascular connections in a gravid female boa 38 days after ovulation—flow towards the transducer is depicted in red, while flow away from the transducer is shown in blue. Through Doppler, it was possible to highlight vascular connections between the mother and the egg membrane in all pregnant females.

**Figure 14 animals-11-03069-f014:**
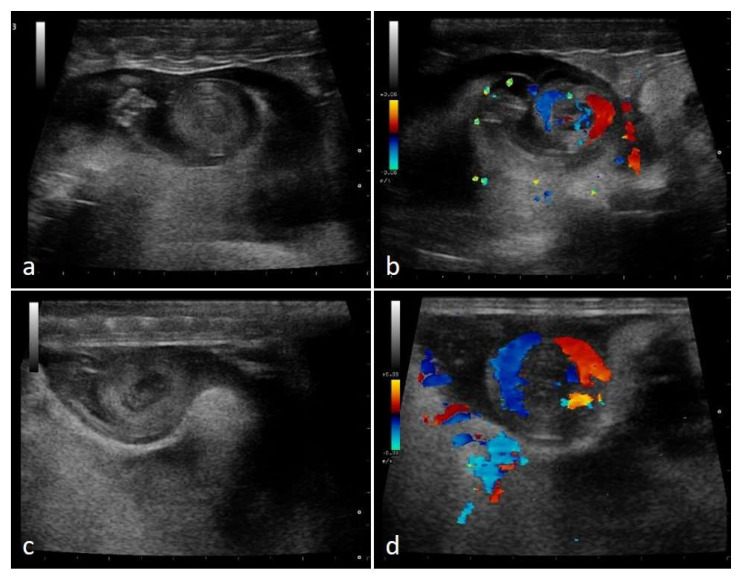
Gravid female boa 52 days after ovulation—flow towards the transducer is depicted in red, while flow away from the transducer is shown in blue. Embryo shows a snail-like appearance (**a**). By using color Doppler, it is possible to visualize vascularization (**b**). Gravid female boa 60 days after ovulation, embryos’ snail-like appearance is observed (**c**). Blood flows are highlighted through Doppler (**d**). Phases recognized in 100% of females who completed gestation.

**Figure 15 animals-11-03069-f015:**
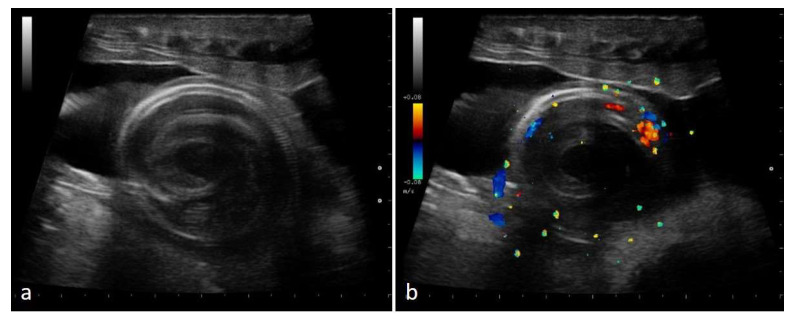
Gravid female boa 77 days after ovulation, 57 days after post-ovulatory shed—flow towards the transducer is depicted in red, while flow away from the transducer is shown in blue. In all females who have given birth to live offspring, embryos showed a spiral appearance, and ribs were well recognizable (**a**). By using color Doppler, it was possible to evaluate vascularization (**b**).

**Figure 16 animals-11-03069-f016:**
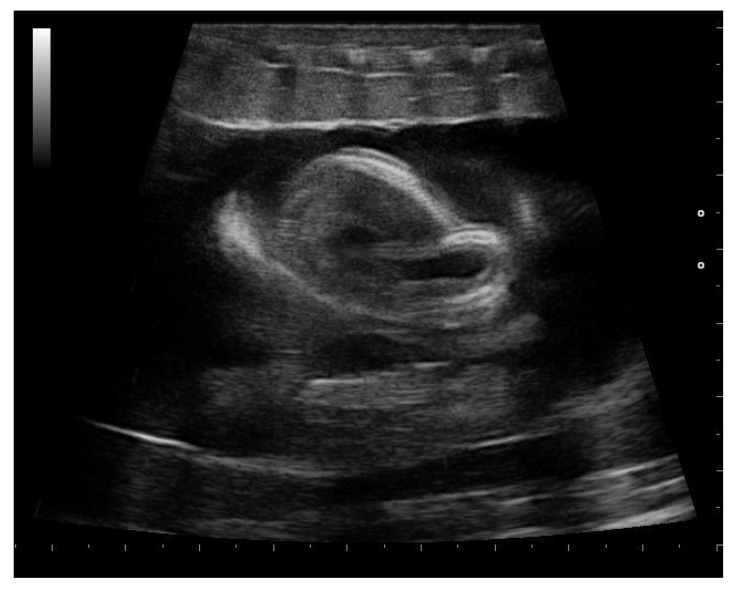
Gravid female boa 4 weeks before parturition. In all females who have given birth to live offspring, embryos moved, and their position varied greatly.

**Figure 17 animals-11-03069-f017:**
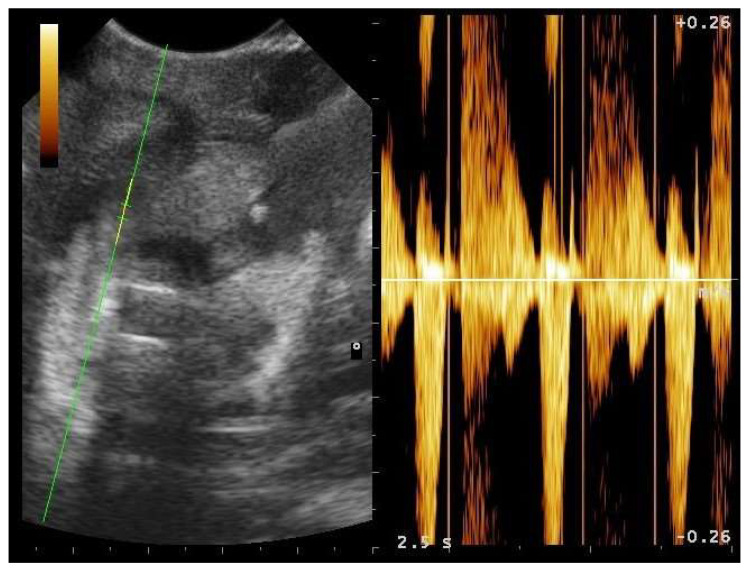
Gravid female boa 33 days before parturition. In all females who have given birth to live offspring, fetal heart was recognized in the first third of the total fetal body length. The sample volume was placed on the embryonic heart, and fetal heart activity was assessed considering heart rate and heart morphology. On the right, the arterial flow is highlighted; in particular, it is possible to observe the systolic peak and the diastolic peak.

## Data Availability

Not applicable.

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
