# Peer review of "Monitoring of the Reproductive Cycle in Captive-Bred Female Boa constrictor: Preliminary Ultrasound Observations"

_animals, 2021, doi:10.3390/ani11113069_

Round 1
Reviewer 1 Report
Dear Authors,
Thank you for submitting this interesting paper on ultrasonography and its uses in monitoring the reproduction in the boa constrictor. In principle, I found the paper to be interesting and useful.
However, there appeared to be some missing information in the work, particularly within the methods and the results. Some of the key information about each of the snakes sampled is missing. For example, it is not clear which snakes were primiparous, and which were not. This potentially influences the results. There are also statistical tests present in the results which are not explained in the methods. As a result, it is difficult to determine what statistical testing was done.
Overall, the results section was the area in greatest need of development. Whilst some useful pictures were provided showing difference stages of embryo development, it is not clear what proportion of snakes showed similar development. In some sections, it is merely stated that 'some' snakes showed this development. As a scientific paper, there needs to be more quantifiable data. What proportion followed development stage x or y? Describe the data so that readers will know what is considered 'normal' according to your data, and what level of deviance was shown between snakes.
Additionally, further consideration should be given to the fact that ultrasonography has been used previously in these snakes - there is a paper dating back to 1993 on this subject in the Boa constrictor. You may want to consider in more detail what makes your study unique.
Author Response
- R1: However, there appeared to be some missing information in the work, particularly within the methods and the results. Some of the key information about each of the snakes sampled is missing. For example, it is not clear which snakes were primiparous, and which were not. This potentially influences the results.
AU: Thank you for the correction. In the materials and methods section, the number of primiparous and pluriparous animals has been specified. Based on your valid consideration, the total number of resorption cases was indicated, and in particular the number of cases of pluriparous and primiparous was specified. We have also highlighted that as regards the ultrasound aspect of follicles and embryos, no differences were found between primiparous and pluriparous.
- R1: There are also statistical tests present in the results which are not explained in the methods. As a result, it is difficult to determine what statistical testing was done.
AU: Thank you for the correction. The use of statistical tests has been specified in the materials and methods section.
- R1: Overall, the results section was the area in greatest need of development. Whilst some useful pictures were provided showing difference stages of embryo development, it is not clear what proportion of snakes showed similar development. In some sections, it is merely stated that 'some' snakes showed this development. As a scientific paper, there needs to be more quantifiable data. What proportion followed development stage x or y? Describe the data so that readers will know what is considered 'normal' according to your data, and what level of deviance was shown between snakes.
AU: Thank you for your observation. We have modified the results section, specifying the proportion of animals that have presented a similar stage of development, compared to those who have deviated from it, in order to clarify what can be considered "normal" by readers. We would like to underline that this is a descriptive and preliminary study, with the aim of evaluating the usefulness of ultrasound in the reproduction of the boa constrictor, highlighting the ultrasound aspect of the reproductive structures and the possibility of non-invasive monitoring during pregnancy. We tried to explain it more clearly at the end of the introduction section. We certainly intend to deepen these issues in a subsequent study, expanding the number of the group of animals studied.
- R1: Additionally, further consideration should be given to the fact that ultrasonography has been used previously in these snakes - there is a paper dating back to 1993 on this subject in the Boa constrictor. You may want to consider in more detail what makes your study unique.
AU: Thank you for the observation. We have tried to deepen this issue in the conclusion section, underlining the strengths of this study and what makes it unique.
Reviewer 2 Report
See references – page 66
Blackburn DG. Discrepant usage of the term “ovoviviparity” in the herpetological literature. Herpetol J 4: 65–72, 1994.
Garcia VC, Almeida-Santos SM. Reproductive cycles of neotropical boid snakes evaluated by ultrasound. Zoo Biology, 2021.
Page 102 - Correctly cite the species name - Boa constrictor
Author Response
Reviewer 2
- R2: See references – page 66
Blackburn DG. Discrepant usage of the term “ovoviviparity” in the herpetological literature. Herpetol J 4: 65–72, 1994.
Garcia VC, Almeida-Santos SM. Reproductive cycles of neotropical boid snakes evaluated by ultrasound. Zoo Biology, 2021.
AU: Thank you for the observation. We tried to improve the introduction and the discussion sections by adding the suggested references. The References section has been accordingly modified.
- R2: Page 102 - Correctly cite the species name - Boa constrictor
AU: Thank you for the correction. The species name has been corrected.
Round 2
Reviewer 1 Report
Dear Authors,
Thank you for submitting a revision of your paper. You have addressed some of the key issues associated with the paper. There are however a couple of key comments to be addressed. One point relates to the use of the images. Currently, it still isn't clear how reflective the images are of the reproductive stages that were seen throughout the study. For example, was it reflective of 100% of all cycles, or did only 50% of snakes show this stage?
The other issue surrounds the normality of the data. As you have used t tests, data should be normally distributed. Did you test for normality? If so, state this in the manuscript.
Author Response
- R: Thank you for submitting a revision of your paper. You have addressed some of the key issues associated with the paper. There are however a couple of key comments to be addressed. One point relates to the use of the images. Currently, it still isn't clear how reflective the images are of the reproductive stages that were seen throughout the study. For example, was it reflective of 100% of all cycles, or did only 50% of snakes show this stage?
AU: Thank you for the observation. Percentages have been inserted in figures captions, in order to clarify how representative they are of the reproductive stages that were seen in the study.
- R: The other issue surrounds the normality of the data. As you have used t tests, data should be normally distributed. Did you test for normality? If so, state this in the manuscript.
AU: Thank you for the correction. Yes, data was tested for normality through the Shapiro-Wilk Normality Test. As they were normally distributed, t test was applied. We have specified this in the manuscript as suggested.
